# Biasing the Future: Gaussian Attention for Sequential Decision-Making

## Abstract

Transformers have emerged as powerful sequence models for offline reinforcement learning (RL), but their reliance on purely self-attention mechanisms can limit their ability to capture fine-grained local dependencies and Markovian dynamics present in many RL datasets. In this work, we introduce a modified Decision Transformer architecture that incorporates a Gaussian-biased masked causal attention mechanism. By augmenting attention scores with a distance-aware bias, the model adaptively emphasizes temporally local relationships while still retaining the ability to capture long-range dependencies through self-attention. Experimental results on benchmark offline RL tasks show that our Gaussian-biased Decision Transformer achieves achieves state-of-the-art performance and notable gains over the standard DT, particularly in environments with strong Markovian structure. This demonstrates the importance of explicitly encoding locality into attention mechanisms for sequential decision-making.

## 1 Introduction

Offline reinforcement learning (RL) addresses the problem of learning policies from fixed datasets without additional interaction with the environment (7). This setting is especially important in domains such as robotics, healthcare, and recommendation systems, where online exploration may be unsafe or impractical (15). Offline RL, though promising, is inherently constrained by the absence of active exploration and its sensitivity to distributional shift; the agent must train exclusively on a fixed dataset $\mathcal{D}$, lacking any mechanism to acquire new transitions that might reveal high-return regions in the state–action space.

Return-Conditioned Supervised Learning (RCSL) provides an effective framework for offline RL (18; 9; 3). In this approach, policy learning is formulated as a conditional prediction problem in which the agent predicts actions from states or histories conditioned on a desired return. This reframing aligns reinforcement learning with supervised learning methods, allowing the use of powerful supervised learning techniques and architectures for policy optimization. Building on this foundation, Conditional Sequence Modeling (CSM) approaches extend the idea by representing trajectories as sequences and modeling them directly. Among these, the Decision Transformer (DT) has become a prominent example, recasting RL entirely as a sequence modeling problem and achieving strong empirical performance in offline RL tasks (3; 22; 21). A more detailed overview of offline reinforcement learning is provided in Appendix A for readers unfamiliar with the topic.

Conditional Sequence Modeling (CSM) approaches benefit significantly from recent advances in sequence modeling, particularly architectures like *MetaFormers*, which provide a powerful and general framework for representing sequences by decoupling the architectural backbone from the specific choice of token-mixing operation (24). Within this framework, sequences such as reinforcement learning trajectories can be modeled using any suitable token mixer while preserving the core architectural components including residual connections, normalization layers, and feedforward networks that ensure stability and expressivity (figure 2).

Building on this general framework, DT leverages the transformer architecture to model trajectories in reinforcement learning as tokenized sequences of returns, states, and actions (figure 1) (3). Specifically, a trajectory is represented as an ordered sequence $(\hat{R}_t, s_t, a_t, \hat{R}_{t+1}, s_{t+1}, a_{t+1}, \ldots)$ where $\hat{R}_t$ denotes the desired return-to-go at time $t$. This sequence is fed into a causal Transformer with self-attention layers as token-mixer that capture long-range dependencies across time steps, enabling the model to learn temporal correlations between past returns, states, and actions. By conditioning the prediction of each action on the preceding context including both the desired future return and past observations, DT learns a return-conditioned policy without explicitly estimating value functions. This design allows DT to directly exploit the powerful sequence modeling and representation learning capabilities of Transformers for decision-making tasks. The core component enabling the Decision Transformer to model trajectories effectively is the *self-attention* mechanism (19).

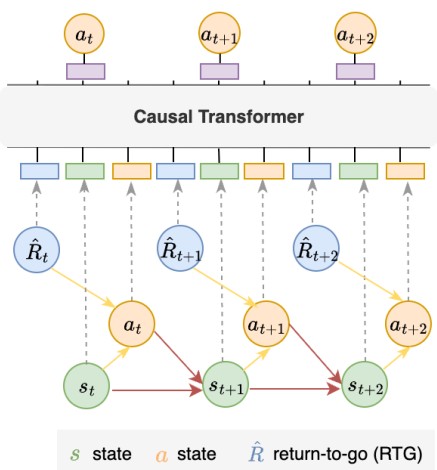

Figure 1: Architecture of the Decision Transformer (DT). Red arrows illustrate the Markov property. Yellow arrows indicate that the selection of each action is influenced by both the current state and the corresponding return-to-go.

Self-attention computes a weighted representation of each token (e.g., returns, states, actions) by attending to all other tokens in the sequence, allowing the model to capture *long-range temporal dependencies* (figure 2). This property relaxes the strict Markovian assumption in reinforcement learning by conditioning decisions on extended historical context rather than only the most recent state. However, the same mechanism can also become a limitation in offline RL. By distributing attention weights broadly across all past timesteps, standard self-attention may dilute the *local*, decision-critical information necessary for predicting the next action, potentially impairing the model's ability to capture short-horizon dynamics that are essential for fine-grained decision-making (10). This limitation arises from the Markovian property of state transitions (figure 1), which dictates that the next state at any given timestep primarily depend on the current state and action rather than the full sequence of preceding states (2). As a result, recent states encode the most critical information for accurate decision-making, whereas distant states generally provide diminishingly relevant information. By distributing attention broadly across all tokens in a trajectory, DT may assign undue weight to temporally distant states, potentially weakening decision-critical signals and impairing the model's ability to exploit the inherent short-term correlations present in offline RL datasets (10). Addressing this discrepancy is crucial for enhancing the sample efficiency and predictive accuracy of sequence-based policy learning models.

Recent approaches have attempted to address the limitation of standard self-attention in offline RL by introducing temporal inductive biases. For instance, the Decision Conformer (DC) (10) leverages convolutional modules to capture local, short-term dependencies, while the Long-Short Decision Transformer (LSDT) (20) combines convolution with attention to balance both short- and long-term interactions. Although effective to some extent, these methods either overemphasize local patterns at the expense of global context (DC) or introduce additional architectural complexity (LSDT), limiting their performance.

Decision ConvFormer (DC) addresses this limitation by replacing DT's attention module with causal convolution filters, explicitly modeling local temporal correlations. DC excels at capturing short-term dependencies and inter-modal relationships with fewer parameters and faster training, but underperforms in environments requiring long-range dependencies, such as non-Markovian or mixed-policy datasets. Long-Short Decision Transformer (LSDT) improves upon this trade-off using a dual-branch architecture, combining a long-term self-attention branch for global dependencies with a short-term dynamic convolution branch for local dependencies. This flexible design enables robust performance across diverse RL tasks and allows modular integration of alternative local modeling strategies.

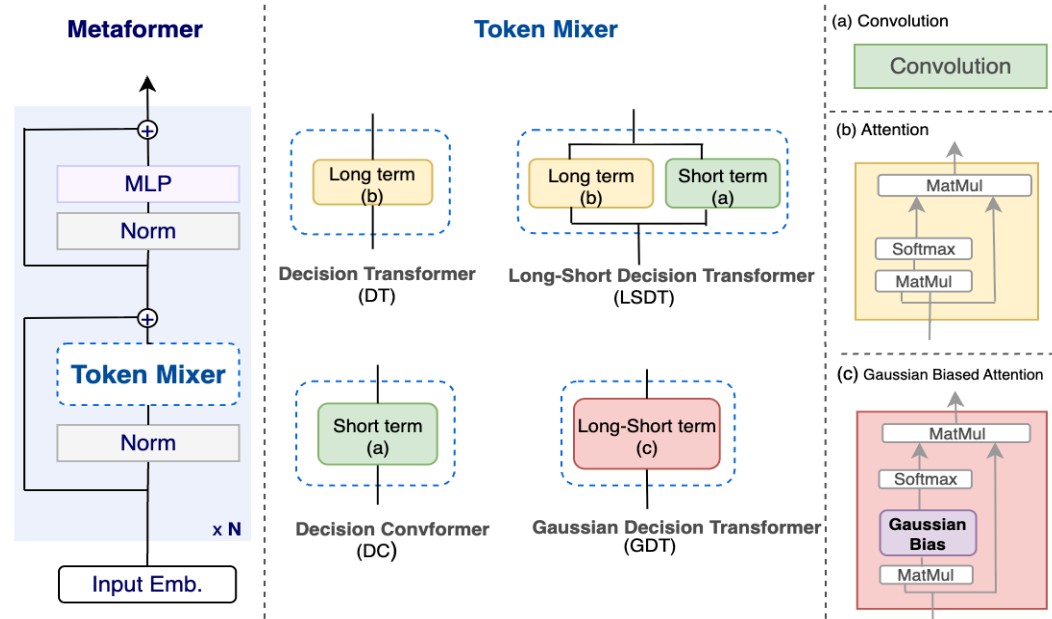

Figure 2: Comparison of different token mixer designs within the Metaformer framework. Decision Transformer (DT) captures long-term dependencies using self-attention. Long-Short Decision Transformer (LSDT) combines self-attention for long-term dependencies with convolution for local information. Decision Conformer (DC) relies purely on convolution for short-term modeling, while our proposed Gaussian Decision Transformer (GDT) introduces a Gaussian-biased attention mechanism to balance local and global context effectively.

Motivated by advances in Natural Language Inference, where Gaussian-biased attention effectively emphasizes neighboring tokens while maintaining global context (17; 4), we introduce the Gaussian-Biased Decision Transformer (GDT). GDT incorporates a distance-aware Gaussian bias into the self-attention mechanism, prioritizing temporally local interactions while preserving long-range dependencies (figure 2). This unifies the strengths of DC and LSDT, enabling stable and efficient modeling of offline RL trajectories with mixed temporal dependencies. Our main contributions are:

- We propose a Gaussian-biased Decision Transformer that explicitly encodes temporal locality into the attention scores via a distance-aware bias. This modification allows the model to focus on recent, decision-critical timesteps while retaining the ability to capture long-range dependencies when needed.

- We provide insights into the Markovian structure of offline RL datasets and show how Gaussian biases align with this structure.

- We demonstrate that GDT outperforms both the state-of-the-art DT-based methods across diverse D4RL tasks, achieving better performance in both continuous control and long-horizon sparse-reward settings.

## 2 GAUSSIAN-BIASED DECISION TRANSFORMER

In this section, we present the Gaussian-Biased Decision Transformer (GDT), which integrates a distance-aware Gaussian bias into the self-attention mechanism to effectively balance local and global temporal dependencies in RL trajectories.

### 2.1 PRELIMINARIES

**Markov Decision Process (MDP):** We consider an MDP (16) defined by the tuple $(\mathcal{S}, \mathcal{A}, P, r, \gamma)$, where $\mathcal{S}$ is the state space, $\mathcal{A}$ is the action space, $P(s' \mid s, a)$ defines the transition dynamics, $r(s, a)$

is the reward function, and $\gamma \in [0, 1]$ is the discount factor that controls the importance of future rewards. The goal of reinforcement learning is to find a policy $\pi(a \mid s)$ that maximizes the expected return $J(\pi) = \mathbb{E}_\pi \left[ \sum_{t=1}^T \gamma^{t-1} r(s_t, a_t) \right]$.

**return-to-go:** We define the *return-to-go* at time $t$ as $\hat{R}_t = \sum_{t'=t}^T r(s_{t'}, a_{t'})$, the sum cumulative reward starting at timestep $t$. This sequence is fed into a causal Transformer, where the prediction of each action $a_t$ is conditioned on the preceding context, including both the desired return and historical observations.

**Context window and tokenization:** At each time step $t$, we construct the context sequence

$$\tau_{t-K+1:t} = \left( \hat{R}_{t-K+1},\, s_{t-K+1},\, a_{t-K+1},\, \ldots,\, \hat{R}_{t-1},\, s_{t-1},\, a_{t-1},\, \hat{R}_t,\, s_t \right), \tag{1}$$

which contains the desired return-to-go, states, and actions over the past $K$ steps up to the current time $t$. The model then predicts the next action $a_t$ conditioned on this context $\tau_{t-K+1:t}$.

Each element of $\tau_{t-K+1:t}$ is linearly projected into a $d$-dimensional embedding (token). Denoting the resulting token vectors in temporal order by

$$x_1,\, x_2,\, \ldots,\, x_{3K-1} \in \mathbb{R}^{1 \times d}, \tag{2}$$

we thus obtain a sequence of $3K - 1$ tokens representing the context at time $t$.

## 2.2 GAUSSIAN SELF-ATTENTION

Given the token sequence $\{x_1, x_2, \ldots, x_{3K-1}\}$ from Eq. equation 2, standard self-attention computes attention weights using the scaled dot-product:

$$\text{Attn}(Q, K, V) = \text{softmax} \left( \frac{QK^\top}{\sqrt{d}} \right) V, \tag{3}$$

where $Q, K, V \in \mathbb{R}^{(3K-1) \times d}$ are the query, key, and value matrices, and $d$ is the embedding dimension. This formulation treats all tokens equally regardless of their temporal distance, which may overemphasize distant and less relevant tokens in sequential decision-making tasks.

**Gaussian locality prior.** To emphasize temporally local information, we introduce a distance-aware Gaussian bias into the attention mechanism. Let $d_{ij} = |i - j|$ denote the temporal distance between tokens $x_i$ and $x_j$. We define the Gaussian prior as

$$g(d_{ij}) = e^{-\frac{d_{ij}^2}{2\sigma^2}}, \tag{4}$$

where $\sigma > 0$ controls the locality strength: smaller $\sigma$ focuses attention on nearby tokens, while larger $\sigma$ allows longer-range interactions.

**Log-space parameterization.** Rather than multiplying by $g(d_{ij})$ directly, we add its logarithm to the attention logits before the softmax:

$$\alpha_{ij} = \text{softmax}_j \left( \frac{QK^\top}{\sqrt{d}} - w\, d_{ij}^2 \right), \quad w = \frac{1}{2\sigma^2}. \tag{5}$$

we introduce a scalar parameter $w > 0$ to control the decay rate: In our implementation, we treat $w$ as a *hyperparameter* and select its value using validation performance, thereby allowing us to balance the emphasis on local versus global temporal dependencies.

**Central token punishment term.** In addition to the Gaussian locality prior, we introduce a *punishment term* $b$ to mitigate over-attention to the current token itself. Following the approach proposed by Guo et al. (8), we apply a negative bias $b \leq 0$ to reduce, but not completely suppress, the self-attention weight. The modified attention formulation is given by

$$\text{Attn}(Q, K, V) = \text{softmax}\left( \frac{QK^\top}{\sqrt{d}} - \big| w\, d_{ij}^2 + b \big| \right) V, \tag{6}$$

where $b \leq 0$ penalizes over-attention at $d_{ij} = 0$. This strategy ensures that information from the current token is still preserved while encouraging the model to attend more to its temporal neighbors as shown in figure 3.

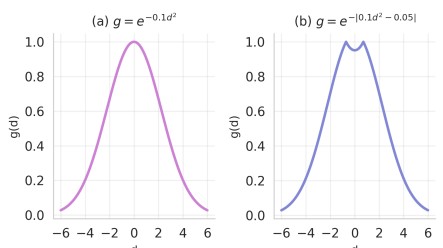

Figure 3: Illustration of Gaussian bias functions: (a) shows the vanilla Gaussian (b) depicts the variant with an additional bias term to penalize central token.

### 2.3 MODEL ARCHITECTURE

GDT adopts the causal Transformer architecture with stacked layers of Gaussian-biased self-attention, layer normalization, and feedforward networks. The overall design remains similar to DT while explicitly encoding temporal inductive biases into the attention mechanism.

### 2.4 TRAINING OBJECTIVE

### 2.5 TRAINING

We train the Gaussian-Biased Decision Transformer (GDT) to predict the next action conditioned on a recent subtrajectory of returns, states, and past actions. Formally, for each trajectory in the offline dataset $\mathcal{D}$, we sample a subtrajectory of length $K$ and process it through all Gaussian-biased Transformer layers described above. The output tokens from the final layer are projected to obtain the action predictions

$$\hat{a}_t = \pi_\theta\big( \hat{R}_{t-K+1:t},\, s_{t-K+1:t},\, a_{t-K+1:t-1} \big), \qquad t = 1, \ldots, K, \tag{7}$$

where $\pi_\theta$ denotes the GDT parameterized by $\theta$. The parameters are optimized by minimizing the mean squared error between the predicted and ground-truth actions:

$$\mathcal{L}_{\text{GDT}} = \mathbb{E}_{\tau \sim \mathcal{D}} \left[ \frac{1}{K} \sum_{t=1}^{K} (a_t - \hat{a}_t)^2 \right]. \tag{8}$$

The implementation details are provided in appendix B

## 3 EXPERIMENTS

Our experiments are designed to evaluate the effectiveness of GDT in offline reinforcement learning tasks. Specifically, we aim to answer the following questions:

- Does GDT improve performance compared to standard Decision Transformer (DT) and its variants?

- Does GDT demonstrate robustness across environments with varying dataset optimality (e.g., medium, medium-replay, and medium-expert datasets)?

- Does GDT improve performance compared to baselines on sparse reward and goal oriented environments?

- Does GDT discover policies that perform better than behavioral policy that generates the dataset?

## 3.1 Experimental Setup

**Environments and Datasets.** We evaluate GDT on the D4RL benchmark suite (5), a standardized collection of offline RL datasets for continuous control. Experiments are conducted in two domains: Gym-MuJoCo and AntMaze, targeting both dense- and sparse-reward settings.

In the Gym-MuJoCo domain, we study three locomotion tasks: Walker2d, Hopper, and HalfCheetah. For each task, we use three datasets: medium, medium-replay, and medium-expert. These datasets differ in terms of data quality and behavioral policies: medium is generated by a partially trained policy, medium-replay combines the full replay buffer collected during training and thus contains more suboptimal trajectories, and medium-expert mixes medium-quality data with expert demonstrations, representing near-optimal behavior. All tasks in this domain feature dense, smooth reward functions that measure the quality of locomotion.

The AntMaze domain evaluates long-horizon, goal-directed navigation with sparse rewards. We use four datasets: umaze, umaze-diverse, medium-diverse, and large-diverse, which differ in maze size and trajectory diversity, requiring trajectory stitching for successful planning.

**Baselines.** We compare GDT against two categories of state-of-the-art methods: *value-based approaches* and *Return-Conditioned Behavioral Cloning* approaches. For value-based approaches, we consider TD3+BC (6), IQL (12), and CQL(14). For CSM approaches, we include the standard Decision Transformer (DT) (3), Q-learning Decision Transformer (QDT) (23), Decision ConvFormer (DC) (10), and Long Short-Term Decision Transformer (LSDT) (20), where LSDT leverages combination of convolution layers and self-attention to capture both long- and short-term temporal dependencies, and DC employs convolutional layers to better extract local features.

**Evaluation Protocol.** Following (5), we report *normalized scores* to enable direct comparison across tasks with different reward scales. For each setting, we average results over 10 evaluation episodes and 3 random seeds.

The normalized score is defined as:

$$\text{Normalized Score} = 100 \times \frac{R_{\text{agent}} - R_{\text{random}}}{R_{\text{expert}} - R_{\text{random}}} \tag{9}$$

where $R_{\text{agent}}$ is the return of the evaluated policy, $R_{\text{random}}$ is the expected return of a random policy, and $R_{\text{expert}}$ corresponds to expert-level performance. A score of 100 indicates expert-level returns. Results are analyzed separately for each domain.

# 4 Results

## 4.1 MuJoCo Continuous Control Tasks

We first evaluate the Gaussian-biased Decision Transformer (GDT) on the MuJoCo continuous control benchmark, comparing it against both value-based and return-conditioned behavioral cloning (RCBC) methods across medium (m), medium-replay (m-r), and medium-expert (m-e) datasets. Table summarizes the normalized scores averaged over three random seeds and ten evaluation episodes.

Across all locomotion tasks, GDT consistently matches or outperforms standard DT and other RCBC baselines. Notably, GDT achieves the highest locomotion mean score (84.7), surpassing LSDT (82.0) and DC (82.2), demonstrating its ability to balance local and global temporal dependencies effectively. For example, on Hopper-medium, GDT reaches 99.1, a substantial improvement over DT (68.4) and even surpassing LSDT (87.2) and DC (92.5). Similarly, on Walker2d-medium-replay, GDT achieves 83.5, the highest among all methods, highlighting its robustness on datasets with mixed-quality trajectories. While value-based approaches like IQL and CQL remain competitive on some datasets (e.g., HalfCheetah-m and Walker2d-m), GDT consistently performs better on sparse or suboptimal datasets where temporal locality is crucial. For instance, on Hopper-medium-replay, GDT reaches 95.9, outperforming all value-based methods, indicating the advantage of modeling local dependencies explicitly in challenging settings. On medium-expert datasets, where trajectories already include near-optimal behavior, most methods perform close to the expert upper bound. Nevertheless, GDT maintains strong performance, achieving 111.9 on Hopper-m-e and 111.2 on Walker2d-m-e, matching or exceeding all baselines. Key insights are as follows:

**Locality Bias Helps on Suboptimal Data:** GDT yields the largest gains on *medium* and *medium-replay* datasets, where local decision-critical signals are essential for accurate action prediction.
**Competitive on Expert Data:** Even when trajectories are near-optimal, GDT performs on par with or better than state-of-the-art baselines, indicating no loss of global context due to the locality bias.
**Best Overall Performance:** Averaged across all tasks, GDT achieves the highest normalized score (84.7), demonstrating its robustness across varying dataset qualities and environments.

| Dataset | Value-Based Method | | | Return-Conditioned BC | | | | | | |
|---|---|---|---|---|---|---|---|---|---|---|
| | TD3+BC | IQL | CQL | DT | ODT | RvS | DS4 | DC | LSDT | GDT (ours) |
| halfcheetah-m | **48.3** | **47.4** | 44.0 | 42.6 | 43.1 | 41.6 | 42.5 | 43.0 | 43.6 | 43.46 |
| hopper-m | 59.3 | 63.8 | 58.5 | 68.4 | 78.3 | 60.2 | 54.2 | 92.5 | 87.2 | **99.1** |
| walker2d-m | **83.7** | **79.9** | 72.5 | 75.5 | 78.4 | 71.7 | 78.0 | 79.2 | **81.0** | **82.8** |
| halfcheetah-m-r | **44.6** | **44.1** | **45.5** | 37.0 | 41.5 | 38.0 | 15.2 | 41.3 | 42.9 | 42.4 |
| hopper-m-r | 60.9 | **92.1** | **95.0** | 85.6 | **91.9** | 73.5 | 49.6 | **94.2** | **93.9** | **95.9** |
| walker2d-m-r | **81.8** | 73.7 | 77.2 | 71.2 | **81.0** | 60.6 | 69.0 | 76.6 | 74.7 | **83.5** |
| halfcheetah-m-e | **90.7** | 86.7 | **91.6** | 88.8 | **94.8** | 92.2 | 92.7 | 93.0 | 93.2 | 92.4 |
| hopper-m-e | 98.0 | 91.5 | 105.4 | **109.6** | **111.3** | 101.7 | **110.8** | 110.4 | **111.7** | **111.9** |
| walker2d-m-e | **110.1** | **109.6** | 108.8 | 109.3 | 108.7 | 106.0 | 105.7 | **109.6** | 109.8 | **111.2** |
| locomotion mean | 75.3 | 76.5 | 77.6 | 76.4 | **81.0** | 71.7 | 68.6 | **82.2** | 82.0 | **84.7** |

Table 1: *Performance comparison on MuJoCo locomotion tasks.* Normalized scores (0-100 scale) are reported, with higher values indicating better performance. Dataset abbreviations: medium (m), medium-replay (m-r), medium-expert (m-e). Each result is averaged over 3 random seeds, and each seed is evaluated on 10 independent episodes. Bold values indicate scores within 95% of the best in each row.

## 4.2 ANTMAZE NAVIGATION TASKS

The AntMaze domain provides a challenging benchmark for evaluating long-horizon planning and decision-making under sparse rewards. Table 2 reports normalized success rates (0–100 scale) for two datasets: antmaze-u and antmaze-u-d. Results are averaged over three random seeds, with each seed evaluated across ten independent episodes.

The proposed Gaussian-Biased Decision Transformer (GDT) consistently achieves the highest success rates across all AntMaze datasets, attaining an average score of 89.1, a significant improvement over both value-based methods and other return-conditioned baselines. On antmaze-u, GDT obtains 90.0, surpassing the best-performing value-based method, IQL (87.1). On the more difficult antmaze-u-d dataset, GDT reaches 88.2, outperforming the strongest baseline, CQL (84.0).

Among value-based methods, CQL performs robustly with an average score of 79.0, while IQL excels on antmaze-u but struggles on antmaze-u-d. Return-conditioned approaches exhibit greater variability: standard DT and ODT underperform on average, whereas DC and LSDT deliver competitive results, suggesting that modeling local dependencies benefits long-horizon planning. Nevertheless, GDT consistently outperforms all baselines, demonstrating the effectiveness of incorporating a Gaussian locality bias for trajectory modeling under sparse rewards.

## 5 ATTENTION ANALYSIS

To better understand how the Gaussian-biased Decision Transformer (GDT) differs from the standard Decision Transformer (DT), we visualize the learned attention weights on the `hopper-medium` dataset (Figure 4). Each panel shows the average attention map across heads and layers for a representative trajectory.

**DT vs. GDT Attention Patterns.** The first two panels reveal a clear difference between DT and GDT. While DT distributes its attention broadly across the trajectory with only weak emphasis on local context, GDT exhibits strong diagonal attention, focusing on recent, decision-critical timesteps.

| Dataset | Value-Based Method | | | Return-Conditioned BC | | | | | | |
|---|---|---|---|---|---|---|---|---|---|---|
| | TD3+BC | IQL | CQL | DT | ODT | RvS | DS4 | DC | LSDT | GDT (ours) |
| antmaze-u | 78.6 | **87.1** | 74.0 | 69.4 | 73.5 | 64.4 | 63.4 | 85.0 | 80.0 | **90.0** |
| antmaze-u-d | 71.4 | 64.4 | **84.0** | 62.2 | 41.8 | 70.1 | 64.6 | 78.5 | 83.2 | **88.2** |
| **Average** | 75.0 | 75.8 | 79.0 | 65.8 | 57.7 | 67.3 | 64.0 | 81.8 | 81.6 | **89.1** |

Table 2: *Performance comparison on AntMaze navigation tasks with sparse rewards.* Normalized success rates (0-100 scale) are reported, with higher values indicating better performance. Dataset abbreviations: umaze (u), umaze-diverse (u-d). Each result is averaged over 3 random seeds, and each seed is evaluated on 10 independent episodes. Bold values indicate scores within 95% of the best in each row. The final row shows the average performance across all AntMaze datasets.

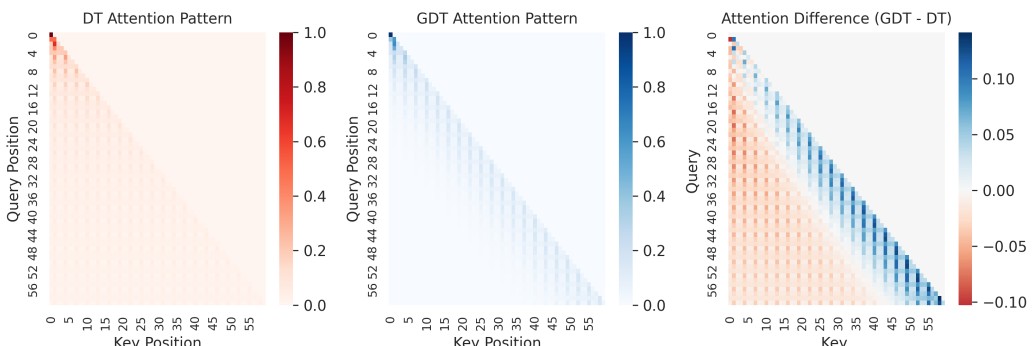

Figure 4: Visualization of attention patterns in the hopper-medium environment for the standard Decision Transformer (DT) and the proposed Gaussian-biased Decision Transformer (GDT). GDT exhibits strong diagonal attention, focusing on recent, decision-critical timesteps, whereas DT distributes attention more diffusely across the trajectory. The difference plot (GDT–DT) highlights that GDT assigns consistently higher weights along the diagonal (blue), reflecting the Gaussian bias toward local temporal dependencies.

This behavior aligns with the Markovian nature of sequential decision-making, where the next state primarily depends on the most recent state and action rather than the entire history.

**Difference Plot (GDT–DT).** The third panel highlights the attention difference between GDT and DT. Blue regions along the diagonal indicate that GDT assigns significantly higher weights to temporally local tokens, while occasional red regions off the diagonal show where DT places relatively more weight on distant tokens. This confirms that the Gaussian bias explicitly enforces locality, preventing over-attention to irrelevant distant states while preserving the ability to capture long-range dependencies when necessary.

The attention analysis demonstrates that GDT learns a structured inductive bias: it prioritizes local temporal information critical for accurate action prediction, yet retains global context for long-horizon reasoning. This explains the substantial performance gains of GDT over DT, especially in environments with suboptimal or noisy trajectories where local decision cues dominate.

## 6 CONCLUSION

We introduced the Gaussian-biased Decision Transformer (GDT), a simple yet effective modification of the standard Decision Transformer (DT) that incorporates a distance-aware Gaussian bias into the self-attention mechanism. By explicitly encoding temporal locality, GDT prioritizes decision-critical recent timesteps while preserving the flexibility of global self-attention.

Our analysis across diverse offline RL benchmarks demonstrates that GDT consistently outperforms DT and several state-of-the-art baselines, particularly in settings with suboptimal or noisy trajecto-

ries where local information is essential. Attention visualizations further reveal that the Gaussian bias induces structured, diagonal attention patterns, aligning model behavior with the underlying Markovian structure of sequential decision-making tasks.

Overall, GDT offers a principled way to integrate inductive biases into attention mechanisms for offline RL, bridging the gap between local temporal reasoning and long-horizon planning. Future work may explore adaptive or learnable locality priors, extending GDT to broader sequence modeling tasks beyond reinforcement learning.

## 7 REPRODUCIBILITY

All code, configuration files, and detailed instructions required to replicate our results is provided in (anonymous).

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

# Appendix

## A  PRELIMINARIES

### OFFLINE REINFORCEMENT LEARNING

Offline Reinforcement Learning (RL) focuses on learning solely from a fixed dataset of transitions $\mathcal{D} = \{(s, a, r, s')\}$, generated by some unknown behavior policy $\pi_\beta$. The goal is to learn a policy $\pi(a \mid s)$ that maximizes the expected return $J(\pi)$ when deployed in the environment, using only the offline dataset (15).

The offline setting introduces unique challenges, prompting the development of a range of methods broadly categorized into *value-based* and *value-free* approaches (15; 21).

**Value-based methods:** These methods aim to learn a value function, typically the state-action value $Q(s, a)$. Once the $Q$-function is estimated, the policy is derived by acting greedily with respect to $Q$. The learning objective minimizes the Bellman error:

$$\mathcal{L}_{\text{Bellman}} = \mathbb{E}_{(s,a,r,s')\sim\mathcal{D}} \left[ \left( Q(s,a) - \left( r + \gamma \, \mathbb{E}_{a'\sim\pi(\cdot|s')}[Q(s', a')] \right) \right)^2 \right]. \tag{10}$$

A key challenge arises from the *distributional shift*: the dataset $\mathcal{D}$ reflects $\pi_\beta$, while the learned policy $\pi$ induces a different distribution. Because the Bellman target involves sampling $a'$ from $\pi$, many such actions may be *out-of-distribution (OOD)* relative to $\mathcal{D}$, producing unreliable $Q(s', a')$ estimates. This *extrapolation error* (7) often destabilizes training and degrades performance.

Two major strategies address this problem:

- **Policy constraints:** Enforce similarity between the learned policy $\pi$ and the behavior policy $\pi_\beta$ (7; 13).
- **Regularization:** Methods such as Conservative Q-Learning (CQL) (14) and Implicit Q-Learning (IQL) (11) penalize overestimated Q-values or impose conservative updates, shaping the policy without explicitly modeling $\pi_\beta$.

**Value-free methods:** These approaches skip explicit value estimation and optimize policies directly, avoiding extrapolation errors altogether. A notable example is the *Decision Transformer (DT)* (3), which reformulates RL as a sequence modeling task using Transformer architectures (19).

DT represents trajectories as sequences of Return-to-Go (RTG), state, action triplets over a horizon $H$ and trains the model to predict the next action given historical context. By leveraging extended temporal dependencies rather than bootstrapped value targets, DT achieves greater stability and is well-suited for modeling long-horizon behaviors (18).

## B  IMPLEMENTATION DETAILS

### B.1  POLICY

Our policy architecture is based on the Decision Transformer framework, employing a GPT-2 style transformer with causal attention. We built upon the open-source implementation available at `https://github.com/nikhilbarhate99/min-decision-transformer`. The hyperparameters used for the model are summarized in Table 3.

For training, we ran $10^5$ gradient steps on the AntMaze domain and $5^4$ gradient steps on the MuJoCo domains. We set the conditioning target return-to-go (RTG) during evaluation to twice the maximum trajectory return observed in the dataset.

Table 3: Hyperparameters of Policy in our experiments.

| Parameter | Value |
|---|---|
| Number of layers | 3 |
| Number of attention heads | 2 |
| Embedding dimension | 128 |
| Activation function | ReLU (1) |
| Batch size | 64 |
| Context length $K$ | 20 |
| Dropout | 0.1 |
| Learning rate | 3.0e-4 |
| w | 0.1 |
| b | -0.05 |

