# OpenReview forum: "Biasing the Future: Gaussian Attention for Sequential Decision-Making"
_ICLR.cc/2026/Conference — ICLR 2026 Conference Withdrawn Submission_

### Official Review · Reviewer_vNpi · 2025-10-21

**Soundness:** 1
**Presentation:** 2
**Contribution:** 1
**Rating:** 2
**Confidence:** 5

**Summary:**

This paper proposes the Gaussian-Biased Decision Transformer, which is a modification of the standard Decision Transformer (DT) architecture used in offline reinforcement learning (RL).

**Strengths:**

The Gaussian bias is a simple, mathematically grounded modification that adds strong inductive priors without increasing model complexity.
The approach can extend to any sequence modeling task requiring local sensitivity—beyond RL, potentially to NLP or time-series forecasting.

**Weaknesses:**

The Gaussian width parameter is manually tuned. A **learnable or adaptive σ** could make the model more flexible across environments.
The paper lacks comparison with state-of-the-art DT-based methods, and the environments compared in this paper are incomplete. For example, in the antmaze environment, this paper only considers the simplest tasks without comparing more challenging tasks.

**Questions:**

1. The abbreviation for "Conditional Sequence Modeling" has already been introduced in the previous paragraph, so there is no need to write out the abbreviation again.
2. The abbreviation for "Decision Conformer" is DC, so why is the abbreviation for "Decision ConvFormer" also DC?
3. typo: “Eq. equation 2”
4. I understand the authors do not want to use multiplication, but why is addition used in Equation 5 rather than subtraction.
5. In Equation 5, the value of $\omega$ is a constant with respect to $\sigma$, but why do the authors mention "select its value using validation performance," which seems to suggest that $\omega$ is a variable that changes during training?
6. In line 84, since the authors mention "recent states encode the most critical information for accurate decision-making, whereas distant states generally provide diminishingly relevant information," why not directly remove the distant information?
7. The DT-based methods compared in the experimental section are not state-of-the-art algorithms. At the very least, to my knowledge, some of the following DT-based methods significantly outperform the method proposed in this paper.

[1] Q-value regularized transformer for offline reinforcement learning

[2] Q-value Regularized Decision ConvFormer for Offline Reinforcement Learning

---

### Official Review · Reviewer_6i1N · 2025-10-30

**Soundness:** 2
**Presentation:** 1
**Contribution:** 2
**Rating:** 2
**Confidence:** 4

**Summary:**

This paper introduces a modified version of the Decision Transformer architecture for the sequential decision-making problem, aiming to balance long-term and short-term information when making decisions. To achieve this, the authors introduce a modification to the standard self-attention layer within the Decision Transformer. The contribution is demonstrated on a set of locomotion tasks from the D4RL benchmark.

**Strengths:**

- The introduction of a bias term within the self-attention layer seems novel.
- The experimental results show promise, improving over the baselines.

**Weaknesses:**

1. The current presentation of the paper is poor: there are many typos, repeated words, and redefined abbreviations in the same section. Here are a few examples:
- Duplicate "achieves" in L21.
- Redefinition of Conditional Sequence Modeling (CSM) in the Introduction.
- L188: "Eq. equation."
- L287: CSM -> Return-Conditioned Behavioral Cloning.
- L94: Decision Conformer -> Decision ConvFormer
- In lines 288-289, the baseline Q-learning Decision Transformer (QDT) is mentioned, but in Tables 1 and 2, the Online Decision Transformer (ODT) is reported.
- What are the RvS and DS4 baselines in Table 1 and 2?

2. Missing related work section, which makes the proposed method poorly situated in the literature.
3. The proposed modification is somewhat incremental; the novelty over prior work [1][2] is not well described.
4. Lack of analysis on key hyperparameters: $w$ used to control the decay rate. How is it effect to the ability of balancing long-term and short-term dependencies?
5. Section 2.4 is missing.
6. Experiments were conducted with 3 seeds, while prior work typically uses 5 seeds (e.g., Decision ConvFormer). This makes the results less reliable.
7. Lack of discussion on the improvement of GDT over previous works such as Decision ConvFormer (DC) [1] and Long Short Decision Transformer (LSDT) [2].

[1] Kim, Jeonghye et al. "Decision convformer: Local filtering in metaformer is sufficient for decision making." ICLR 2024.\
[2] Wang, Jincheng et al. "Long-short decision transformer: Bridging global and local dependencies for generalized decision-making." ICLR 2025.

**Questions:**

1. What is the effect of key hyperparameters on GDT's performance?
2. In the AntMaze navigation domain, how does GDT compare to DC and LSDT in antmaze-mediumplay and antmaze-medium-diverse?
3. What is the computational cost of GDT compared to DC and LSDT?

---

### Official Review · Reviewer_sSTV · 2025-10-31

**Soundness:** 3
**Presentation:** 3
**Contribution:** 2
**Rating:** 2
**Confidence:** 4

**Summary:**

This paper proposes the Gaussian-biased Decision Transformer, which incorporates a Gaussian locality bias into the self-attention mechanism. The Gaussian bias adjusts attention logits based on temporal distance, emphasizing nearby timesteps while retaining the ability to capture long-range dependencies. Experiments on Mujoco and AntMaze benchmarks show improvements over prior Decision Transformer variants and value-based baselines.

**Strengths:**

- Proposes a simple, generalizable Gaussian bias mechanism
- Antmaze experiments show meaningful improvements over DT baselines
- The paper is clearly presented

**Weaknesses:**

- The Mujoco experiments in Table 1 show that the performance is only slightly better on average and in many environments many methods are bolded which seems to indicate the improvement is not significant
- The contribution is incremental since local attention bias is not novel

**Questions:**

See weaknesses

---

### Note · Authors · 2025-12-10

I have read and agree with the venue's withdrawal policy on behalf of myself and my co-authors.